# Endopolyploidy Variation in Wild Barley Seeds across Environmental Gradients in Israel

**DOI:** 10.3390/genes12050711

**Published:** 2021-05-10

**Authors:** Anna Nowicka, Pranav Pankaj Sahu, Martin Kovacik, Dorota Weigt, Barbara Tokarz, Tamar Krugman, Ales Pecinka

**Affiliations:** 1Centre of the Region Haná for Biotechnological and Agricultural Research, Institute of Experimental Botany, Czech Academy of Sciences, Šlechtitelů 31, 779 00 Olomouc, Czech Republic; nowicka@ueb.cas.cz (A.N.); sahu@ueb.cas.cz (P.P.S.); kovacik@ueb.cas.cz (M.K.); 2The Franciszek Górski Institute of Plant Physiology, The Polish Academy of Sciences, Niezapominajek 21, 30-239 Krakow, Poland; 3Global Change Research Institute of the Czech Academy of Sciences, Bělidla 986/4a, 603 00 Brno, Czech Republic; 4Department of Genetics and Plant Breeding, Poznan University of Life Sciences, 11 Dojazd St., 60-632 Poznan, Poland; dorota.weigt@up.poznan.pl; 5Department of Botany, Physiology and Plant Protection, Faculty of Biotechnology and Horticulture, University of Agriculture in Krakow, Al. 29 Listopada 54, 31-425 Krakow, Poland; barbara.tokarz@urk.edu.pl; 6Institute of Evolution, University of Haifa, Abba Khoushy Ave. 199, Haifa 3498838, Israel

**Keywords:** Endoreduplication, endosperm, *Hordeum vulgare* ubsp. *spontaneum*, seed development, super cycle value

## Abstract

Wild barley is abundant, occupying large diversity of sites, ranging from the northern mesic Mediterranean meadows to the southern xeric deserts in Israel. This is also reflected in its wide phenotypic heterogeneity. We investigated the dynamics of DNA content changes in seed tissues in ten wild barley accessions that originated from an environmental gradient in Israel. The flow cytometric measurements were done from the time shortly after pollination up to the dry seeds. We show variation in mitotic cell cycle and endoreduplication dynamics in both diploid seed tissues (represented by seed maternal tissues and embryo) and in the triploid endosperm. We found that wild barley accessions collected at harsher xeric environmental conditions produce higher proportion of endoreduplicated nuclei in endosperm tissues. Also, a comparison of wild and cultivated barley strains revealed a higher endopolyploidy level in the endosperm of wild barley, that is accompanied by temporal changes in the timing of the major developmental phases. In summary, we present a new direction of research focusing on connecting spatiotemporal patterns of endoreduplication in barley seeds and possibly buffering for stress conditions.

## 1. Introduction 

Cultivated barley (*Hordeum vulgare* subsp. *vulgare*, 2n = 2x = 14) was domesticated about 10,000 years ago from its progenitor wild barley (*H. vulgare* subsp. *spontaneum*, 2n = 2x = 14). Barley belongs to the group of “Neolithic founder crops”, and was one of the first cereals that became a pillar of food and feed for ancient societies [1]. The Fertile Crescent is the center of barley domestication, distribution, and diversity [1]. A recent archegenomic study performed on ancient DNA of 6000 years-old barley grains excavated at a cave in the Judean Desert in Israel, narrowed its domestication region to the Upper Jordan Valley [2]. Subspecies *spontaneum* is distributed from eastern North Africa, through the Middle East to India and west China [1]. It constitutes an important annual element of open herbaceous and park-like vegetation [3]. Wild barley natural habitats are characterized by wide ecogeographical diversity caused mostly by contrasting climatic and topographic conditions within the East Mediterranean region. This is reflected by its phenotypic and genetic heterogeneity [4]. During the last century, wild barley was collected all over its distribution area and seed samples are stored and maintained in ex situ gene-banks [3,4]. While domestication and modern plant breeding have reduced the genetic diversity of cultivated barleys, the stocks of subsp. *spontaneum* form a major source for variability, novel genes, and alleles for barley breeding [5,6]. For example, wild barley was found to be an important source of resistance to biotic and abiotic stresses, including multiple diseases [7], tolerance to cold [8], drought [9], and salt [10].

Cereal grain development includes three major phases, characterized by different cellular and physiological events (Figure 6a–c quoted from [11]). Phase I starts with double fertilization and passes smoothly into the cells/nuclei proliferation; phase II comprises differentiation of embryo and endosperm tissues, and seed mass gain by the accumulation of storage compounds; phase III corresponds to seed maturation, weight reduction by desiccation, and an onset of dormancy. These phases partially overlap with three morphological caryopsis growth stages named water, milk, and dough, respectively [12]. 

Cereal grain consists of three major compartments: multilayered seed maternal tissues (SMTs; nucellar projection, pericarp plus seed coats), endosperm, and embryo. The pericarp (diploid, 2x) is derived from the ovary wall and adheres strongly to the seed coats of the ovule [13]. Within the first days after pollination (DAP), the pericarp serves to protect and support the growing endosperm and embryo by starch deposition and photosynthesis in cultivated barley [14,15]. During double fertilization, one sperm nucleus fuses with the egg cell nucleus and gives rise to the diploid embryo (2x), while the second sperm cell nucleus fuses with the diploid central cell to form a triploid endosperm (3x) with the peculiar genetic constitution of one paternal and two maternal genomes. Endosperm nuclei first form syncytium (a.k.a. coenocyte) and later endosperm cellularizes and differentiates into five specialized tissues: the central starchy endosperm (CSE), the sub-aleurone layer (SAL), the aleurone layer (AL), the basal endosperm transfer layer (BETL), and the embryo-surrounding region (ERS) [16]. Endosperm protects and nourishes the embryo. It is the main caryopsis part accumulating primarily sugars and proteins [13,16]. The cereal kernel is covered by hulls that consist of the lemma, palea, and glumes of maternal origin and which remain tightly attached to the grain even after ripening [17]. 

Both SMTs and endosperm tissues undergo genetically controlled endoreduplication during seed development in cultivated barley [11]. Endoreduplication (a.k.a. endopolyploidization) occurs via the endocycle and is a variant of the cell cycle, in which cell nuclei increase their ploidy through repeated rounds of replication without cell divisions (reviewed, e.g., in [18,19]). To unravel the mechanism involved in the switch from a mitotic cell cycle to an endocycle many cyclin-dependent kinases (CDKs), their cyclin partners, CDK inhibitors (e.g., WEE1), and retinoblastoma-related (RBR) proteins have been studied [20]. Despite many efforts, the knowledge about the molecular control of endoreduplication is fragmentary. In most Angiosperms, endoreduplication is common in specialized cells producing secondary metabolites and/or as a means to accelerate cell expansion of specific tissues [20]. Also, various abiotic and biotic factors affect the endopolyploidy level of cells and tissues [21,22]. For instance, salinity or the absence of light stimulates extra endocycles in different Arabidopsis organs [23,24]. Endoreduplication can also be triggered upon symbiotic [25] and also pathogenic [26] plant-microbe interactions. In contrast, endopolyploidization can be repressed by both very high and very low temperatures [22] or drought [27]. 

The development of cereal seeds would not be possible without programmed cell death (PCD). In phase I, maternal tissues, i.e., components of the embryo sac, nucellus, nucellar projection, seed coats, and pericarp undergo a progressive degeneration by PCD [28,29]. During phases II and III, mainly two endosperm parts: ESR and CSE undergo cell death, but the cells remain intact in the mature grain and their contents will not be remobilized until germination. Finally, the mature grain contains mainly dead material, where only the embryo, BETL, and AL tissues remain alive [30,31,32]. 

Wild barley is a generalist abundant across diverse habitats ranging from the mesic Mediterranean meadows to the xeric southern habitats and even penetrating the central Negev desert in Israel. Such environmental heterogeneity can be a direct driving force for adaptation [4]. The main objective of this study was to investigate the dynamics of endoreduplication in seed tissues of wild barley originating from mesic, semi-mesic, semi-xeric, and xeric ecogeographic sites of Israel. For this purpose, we measured the DNA contents in diploid seed tissues (embryo and maternal tissues) and triploid endosperm using flow cytometry. We calculated the proportion of nuclei with different DNA contents and estimated the level of endoreduplication with a new formula called the super cycle value (SCV) [11]. For a better understanding of the dynamics of processes associated with wild barley grain development, we also monitored the morphology of developing seeds and performed Evans blue cell death assay. We found that wild barley accessions originating from the xeric environments have on average higher proportion of endoreduplicated nuclei in seed tissues, and tend to have a higher SCV index. This indicates the impact of harsh conditions on endoplyploidization. A comparison of wild and cultivated barleys reveals a higher endopolyploidy level in the endosperm of wild barley that is accompanied by temporal changes in the timing of the major developmental phases.

## 2. Materials and Methods 

### 2.1. Plant Materials and Growth Conditions 

Ten wild barley (*H. vulgare* subsp. *spontaneum*) accessions originating from Israel were used in this study. Seeds were obtained from the Institute of Evolution Wild Cereal Gene Bank (ICGB) at the University of Haifa, Israel, and Leibniz Institute of Plant Genetics and Crop Plant Research (IPK), Gatersleben, Germany. The ICGB accessions were named based on the seed collection sites and the type of environment (Figure 1; Table 1). Three accessions originated from typical xeric (x) environments: Machtesh Gadol (MGx), Mehola (MHx), and Wadi Qilt (WQx); three from mesic (m) environments: Rosh Pinna (RPm), Tel Hai (THm), and Zefat (ZFm), and one accession from Bar Giyyora represented semi-mesic environment (BGsm). Two accessions originated from Nahal Oren (NO) Canyon, also named “Evolution Canyon” [32]. The first NO accession was collected from the North-facing slope (NFS) representing a mesic environment (NOm), and the second from the South-facing slope (SFS) belonging to the semi-xeric group (NOsx). In brief, the xeric environment is characterized by low annual rainfall and high temperatures, and mesic by high annual rainfall and lower temperatures. Differences between environments are mainly reflected at seed development time during March to April. The environmental conditions at BG are regarded as semi-mesic due to the high rainfall and dry environment in the Judean mountains. The SFS is regarded as semi-xeric due to higher solar radiation as compared with the NFS Nahal Oren. The IPK accession HS584 carries the gene bank name HOR 12560, and the exact site of the collection is unknown. 

Also, published data [11] from six cultivars (cv.) of two-rowed spring barley (*H. vulgare* subsp. *vulgare*): Betzes (PI 129430), Compana (PI 539111), Golden Promise (GP; PI 343079), Ingrid (PI 263574), Klages (CIho 15478) and Mars (PI 599629) and three additional cv. of six-rowed spring barley: Glacier (CIho 6976), Mars (CIho 7015) and Morex (BCC 906) were used for comparison.

Grains were stratified in the dark at 4 °C for 48 h, evenly spread on wet filter paper in a Petri dish, covered with a lid, and germinated at 25 °C for 3 days in the dark. Germinating kernels were planted into 5 cm × 5 cm peat pots with a mixture of soil and sand (2:1, *v*/*v*) and grown in an air-conditioned phytochamber with a long day regime (16 h day with 20°C and 200 μmol m^−2^ s^−1^ light intensity; 8 h night with 16 °C; 60% humidity). After 10 days, wild barley plants were placed into the vernalization chamber (short-day regime; 8 h day with 4 °C, light intensity 200 µmol m^−2^ s^−1^; 16 h night with 4 °C; humidity 85%) for three weeks. Ten-day-old cultivated barley plants and 31-day-old wild barley plants were transferred into the 12 cm × 12 cm pots filled with the above-described soil mixture and grown under long-day conditions. For each accession five plants were grown. Day of pollination (DOP) was monitored using the morphology of stigma and anthers according to the Waddington scale (W10) [33] as we described previously [11,34]. In brief, the spikelets at DOP were characterized by extended hulls, widely branched stigma, and the presence of pollen grains on stigmatic hairs. Seeds were collected from the center of the spikelet at two- and four-day intervals, starting from 4 until 24 days after pollination (DAP). For this experimental setup, in total seven-time points were examined (i.e., 4, 6, 8, 12, 16, 20, 24 DAP). For three accessions, HS584, RPm, and BGsm collecting the seeds were extended up to 48 DAP (additional six collection points: 28, 32, 36, 40, 44, 48 DAP). Mature dry seeds (called ‘dry seeds’ latter in the text) were harvested around 60–65 DAP from fully dried mother plants, cleaned, and stored first ~30 days at 20 °C, then ~60 days at 4 °C, both in darkness. The analysis was performed after 90 ± 5 days after harvesting the seeds. During collecting the seed from mother plants, kernels were at the hard-dough phase of barley grain development (87–89 stages according to [12]). It means that grains were dry and cannot be squeezed out. The maximum dry seed section area was reduced by approximately 30–40% as compared to 20–28 DAP seeds (Appendix A). Hulls had yellow color.

### 2.2. Analysis of Nuclear DNA Content and Calculation of the Super Cycle Value (SCV)

Nuclear DNA contents were estimated using flow cytometry (FCM). For each time point, five to six individual seeds, freshly collected from one spike were analyzed. The measurements were repeated three times on different days using seeds harvested from different mother plants and keeping the same time of day for analysis. The isolation of nuclei and estimation of nuclear DNA content was performed as previously described [11]. Briefly, seeds directly after harvesting were cleaned by removing hulls using tweezers. Then, single seeds were immediately homogenized with a razor blade in a Petri dish containing 500 µL of Otto I solution (0.1 M citric acid, 0.5% Tween 20). The crude suspension was filtered through 50 µm nylon mesh (Sysmex-Partec) and stained around 15 min with 1 mL of Otto II solution (0.4 M Na_2_HPO_4_·12H_2_O) supplemented with 2 µg mL^−1^ DAPI (4′,6-diamidino-2-phenylindole). Nuclei samples were analyzed using either a CyFlow Space or a Partec PAS I flow cytometers (Sysmex-Partec, Muenster, Germany), both equipped with UV-led diode lamps. For calibration of the cytometers, the optics were adjusted using calibration beads (A7304, Invitrogen, Carlsbad, CA, USA) until the coefficient of variation (CV) reached <2%. At least 5000 particles were acquired per sample, using a log3 scale. Histograms were evaluated by the FloMax software (Sysmex-Partec, Muenster, Germany).

To estimate amount of endoreduplication, we used super cycle value (SCV) [11]. In SCV, 8C in the diploid and 12C in the triploid tissues were considered as the first levels of endopolyploid nuclei. Our rationale is, that it is not possible to unambiguously distinguish by FCM whether a given 4C (or 6C nucleus in endosperm) nucleus just entered endoreduplication or will mitotically divide [36]. For diploid tissues SCV = ((n 2C × 0) + (n 4C × 0) + (n 8C × 1) + (n 16C × 2))/(n 2C + n 4C + n 8C + n 16C), and for triploid endosperm SCV = ((n 3C × 0) + (n 6C × 0) + (n 12C × 1) + (n 24C × 2))/(n 3C + n 6C + n 12C + n 24C), n = number of counts per given C-value content.

### 2.3. Determination of Seed Morphology Parameters

Analysis of dry seed morphology parameters was performed in three biological replicates, each with at least 20 seeds collected from four to five spikes of different plants. Dry kernels were peeled off, weighed with an analytical scale (Sartorius, Göttingen, Germany), and photographed using a SZX16 binocular microscope (Olympus, Tokyo, Japan) bonded with a Regita 1300 QImaging camera and QCapture ×64 software (Olympus). Seed length and width were measured using ImageJ calibrated with internal size control. Seeds from 4 to 48 DAP and dry seeds were peeled off and cut with a razor blade along the longitudinal and transverse axis. At least 20 individual seeds were photographed as described above using a binocular microscope. Hulled seeds that possessed awns were photographed with a D5600 (Nikon, Tokyo, Japan) digital camera equipped with an 80 mm Nikkor objective. All photo-matrix were composed of separately taken photos of individual seeds and merged in Adobe Photoshop CS5 (Adobe Inc., San Jose, CA, USA). 

### 2.4. Cell Death Assay by Evans Blue Staining 

Seeds from 4 to 48 DAP and dry seeds were peeled off and cut with a razor blade along the longitudinal and transverse axis. At least 20 individual seeds bulked from four to five spikes of different plants were stained in 0.1% (*w*/*v*) Evans blue (314-13-6, Sigma-Aldrich, St. Louis, MO, USA) for 2 min. Stained sections were washed twice for 10 min with distilled water [31]. Transverse and sagittal sections of samples were analyzed with an SZX16 binocular microscope (Olympus). Images were captured with a Regita 1300 QImaging camera and QCapture ×64 software (Olympus) using the same settings and proceeded in Adobe Photoshop CS5 (Adobe Inc.).

### 2.5. Statistical Analysis

All data after testing for normal distribution were examined by one- or two-way analysis of variance (ANOVA), after which post hoc comparison was performed using Duncan’s multiple ranges (*p* ≤ 0.05) test. Data expressed as percentages were first transformed using arcsine transformation. Principal component (PC) analysis was used to analyze relations between variables. Statistical analyses were performed in Statistica v. 12 (Stat Soft Inc., Tulsa, OK, USA), Minitab v. 18 (Minitab, LLC, State College, PA, USA) or RStudio programs.

## 3. Results

### 3.1. Variation in Mature Dry Seed Morphology of Wild Barley

We used samples from nine geographically distant sites from North to South Israel along with the aridity gradient (Figure 1b; Table 1). Also, we included a commonly used gene-bank accession of wild barley named HS584 from an unknown origin in Israel. The wild barley accessions varied as to their mature dry seed weight, length, width, and awn length (Figure 2a–d; Appendix A). For example, the TKW in wild barleys ranged from 14.2 g (MHx) to 40.5 g (NOm). We noted that seeds of wild barley were longer than those of cultivars (wild barley seed length ≥ 9 mm, cultivars seed length ~7–8 mm) (Appendix A). However, only slight differences between wild barleys appeared for seed width (Appendix A). Seeds of wild barley accessions had longer awn. We also noted an intraspecific variation with xeric accessions having shorter awns than the mesic ones (Appendix A). ANOVA results showed that the values of observed variables between wild barley depended on the accession (genotype), and except for seed length, also from the type of environment (Figure 2d). Using these seed phenotypic data, we performed principal component (PC) analysis (Figure 2c; Appendix A). However, this analysis did not reveal any specific group. We noted that one xeric accession MHx varied from the rest of the wild barleys. In addition, barley cultivars were separated from wild accessions. 

Collectively, these data show a phenotypic variation of wild barley seeds. The shortened seed awn length is the most pronounced feature differentiating xeric barley accessions.

### 3.2. Variation in Endoreduplication Dynamics in Developing Wild Barley Seeds

#### 3.2.1. Diploid Seed Tissues

We used whole peeled seeds (hulls were manually removed) to study the degree of endopolyploidy in the seeds of wild barley. We measured C-values of diploid nuclei from the embryo (EMB) and seed maternal tissues (SMTs, containing: nucellar projection, pericarp, and seed coats) and of triploid nuclei fraction represented by endosperm (END) (Appendix A). These measurements were performed for a period 4–24 DAP and then in dry seeds (Results for endosperm are presented in the next subsection numbered 3.2.2.). Diploid seed tissues contained 2C and 4C nuclei representing G1 and G2 phases of the cell cycle, and 8C and 16C endoreduplicated nuclei originating from one and two endocycles, respectively (Appendix A).

We found that all wild barleys contained similar amounts of 2C and 4C nuclei, each oscillating between 40% to 50%, with a minimal amount ≤ 10% of endoreduplicated nuclei at 4 DAP. Up to 12 DAP the number of endopolyploid nuclei increased to reach the maximum, i.e., 10–24% for 8C and 5–11% for 16C. After 20 DAP, the 2C nuclei fraction increased, while the proportion of 4C, 8C, and 16C was gradually reduced. Finally, in mature dry seeds, 2C nuclei amounted to around half (50–60%), 4C nuclei around 30%, and 8C and 16C nuclei < 20% (Figure 3a; Appendix A). AVOVA results showed that the values of these variables depended on both the type of environment and DAP and the interaction between these two factors (Figure 3c). 

To estimate the degree of endoreduplication, we calculated the SCV parameter (Figure 3b; Appendix A). At 4 DAP, a very low SCV of ≤0.09 was observed for all accessions. From 6 DAP onwards, the SCV increased to reach the peak at 12–24 DAP depending on the genotype. The highest SCV of 0.42, appeared in the two xeric accessions MGx and WQx, at 16 and 24 DAP, respectively. Both accessions originate from the most southern collection sites (Figure 1). Similar to the previous observation, the values of these variables depended on the environment type and DAP, as well as the interaction between these two components (Figure 3c). The SCV curve for HS584 had a very smooth profile without any abrupt changes between neighborhood time points, and resembled the THm SCV line (Figure 3b).

Taken together, these data show endoreduplication variation in developing embryos and/or SMTs of wild barley seeds. The most southern xeric accessions show a tendency for a higher endopolyploidy level.

#### 3.2.2. Triploid Endosperm Tissues

Endosperm seed tissues contained four populations of nuclei, where 3C and 6C values reflected G1 and G2 phases of the mitotic cell cycle, and 12C and 24C nuclei resulted from one and two endocycles, respectively (Appendix A). We calculated the frequencies of individual C-values in all ten wild barley accessions up to 24 DAP, and then in dry seeds (Figure 4a; Appendix A). The inter-accession differences in endosperm C-values were striking already from the beginning of seed development. For instance, the frequency of 3C nuclei ranged from 50% to 80% (MHx vs. MGx, respectively), and 6C nuclei from 14% to 40% (inversely MGx vs. MHx, respectively) at 4 DAP. Only at this time point, all accessions contained a similar amount of endoreduplicated nuclei (≤9%). From 6 to 24 DAP, the amount of 3C decreased approximately two times (from ~60% to ~25%), the fraction of 6C nuclei maintained a constant level (around ~30%), and the amount of 12C and 24C nuclei continuously increased up to 50% for MHx, MGx and WQx (all xeric accessions). In dry seeds, the fraction of 3C nuclei ranged from 31% to 47% (HS584 and NOm, respectively), 6C from 37% to 57% (NOsx vs. MHx, respectively), and endoreduplicated nuclei from 12% to 22% (NOsx vs. HS584, respectively). ANOVA results showed that the values of these variables depended on the environment type or DAP, but not the interaction between these two factors (Figure 4c).

At 4 and 6 DAP, the SCV corresponded to ~0.10. From 8 DAP, the SCV started to increase to reach the peak at 12–24 DAP. During this period, xeric accessions showed generally higher SCV. For example, it was 0.78 for MHx (16 DAP), 0.61 for MGx (16–24 DAP), and 0.66 for WQx (20 DAP). In turn, accessions from the mesic environments showed a slightly lower SCV peak, ranging from 0.47 to 0.63 for RPm (20–24 DAP) and ZFm (16 DAP), respectively. The semi-mesic BGsm and semi-xeric NOsx accessions reached the maximum SCV of 0.60 (20–24 DAP) and 0.54 (6 DAP), respectively (Figure 4b; Appendix A). ANOVA analysis revealed that the values of these variables were both environment- and DAP-, but not additively, dependent (Figure 4c). For HS584, the endosperm SCV profile was the most similar to THm and NO (Figure 4b). 

Collectively, these results demonstrated that wild barley accessions reached the peak of endosperm endoreduplication at 12–24 DAP, and endopolyploidy level tended to be higher in xeric accessions.

### 3.3. Comparison of Endoreduplication Dynamics in Developing Seeds of Wild and Cultivated Barley 

Finding the differences between wild barley accessions, raised the question of whether it differs from cultivated barley. Therefore, we compared the data from the wild and the cultivated barley [11]. 

To provide a representative picture, we calculated the mean SCV at different DAP for diploid tissues (embryo and SMTs) and triploid endosperm for all ten wild barley accessions and nine barley cultivars (Figure 5a; Appendix A). The cultivars were represented by six two-rowed [11] and three six-rowed genotypes (Appendix A). We performed ANOVA to investigate the influence of two parameters: type of the sample (wild barley vs. cultivars) and age of the seeds (Figure 5a). For diploid tissues, ANOVA revealed differences depending on DAP and on the interaction of the sample type and DAP. Both wild and cultivated barleys achieved the highest mean SCV at 12 DAP (0.32—cultivars; 0.33—wild barley). The mean SCV for these two types varied significantly at 4–8 DAP, 20–24 DAP and in dry seeds.

For endosperm tissues, ANOVA revealed dependency of the SCV values on the sample type and DAP, and the interactions between these two factors. Mean SCV for endosperm tissues was higher for wild barley. In wild barley, SCV peaked at 16–20 DAP reaching the value ~0.55. Cultivated barley reached a sharp peak at 16 DAP with SCV ~0.46. The mean SCV for these two types varied significantly at 16–24 DAP and in dry seeds (Figure 5a).

To gain insight into the SCV relationships among the wild and cultivated barley, we performed PC analysis (Figure 5b). The first component (PC1) grouped samples based on DAP and showed similarity between individual experimental points of diploid tissues and endosperm development. For the mix of embryo/SMT nuclei, the SCV analysis revealed two groups: (i) 4 to 8 DAP and (ii) after 16 DAP. Similar sample distribution occurred for endosperm, excluding dry seed sample separated from the rest of time points. The second component (PC2) displayed the associations between the genotypes. The SCV data revealed two groups, the first formed by wild barley accessions and the second by barley cultivars. 

Taken together, these data highlight the large inherent variation between wild and cultivated barley. Interestingly, the level of endoreduplication in endosperm tissues is higher in wild barley.

### 3.4. Morphological and Cellular Changes during 48 Days of Wild Barley Seed Development

The need to explain the reasons for wild vs. cultivated barley endopolyploidy variation, inspired us to extend wild barley seed analysis in a broader experimental context and time. First, we monitored the dynamics of seed growth for HS584 from 4 to 48 DAP, and in dry seeds (Figure 6a; Appendix A). At 4 DAP, SMTs constituted the dominant part of the seed. Both sagittal and transverse seed plans showed endosperm expansion, which accelerated seed growth from 6 DAP onwards (Appendix A) and wild barley seeds reached the maximum growth (whole seed sagittal section areal ~20 mm^2^) at DAP 24. Interestingly, the endosperm changed color from green to gray from DAP 32, and the desiccation started to be visible from DAP 40. The most intense growth of the embryo occurred around DAP 16 (Appendix A). 

Next, we analyzed cell death which is a crucial cellular process during cereal seed development. To detect viable and non-viable cells, we performed Evans blue staining (Figure 6b). The stain penetrates the intracellular spaces of dead tissues and dyes them blue. Cell death followed a specific pattern in developing wild barley seeds. We detected regions of blue staining in the top (seed brush) and bottom parts of SMTs, but not in the longitudinal elongation zone from 6 DAP onwards. In endosperm, very weak blue signals appeared in the central part at 12 DAP, and the area of staining and color intensity increased over time. AL was the only endosperm tissue free of staining at the end of seed development (Appendix A). No staining was observed in the embryo at any stage of seed development. 

Finally, we continued the measurement of nuclear C-values in seed tissues up to 48 DAP for the HS584, RPm, and BGsm accessions (Figure 6c; Appendix A). For diploid seed tissues, the SCV profile differed between accessions. HS584 contained two peaks, at 12 and 32 DAP, reflecting probably the accumulation of endoreduplicated nuclei in SMTs and embryos, respectively. The RPm genotype had one symmetrical peak of endoreduplication from 6 to 28 DAP, which contrasted to very irregular peak of BGsm (Appendix A). The SCV curve had a single broad peak profile for three studied accessions in endosperm tissues (Figure 6c). The differences concerned its width, reflecting the shifts between the start and end of the endopolyploidy period. Interestingly, the transition between low SCV at 48 DAP and its higher value in dry seed was very clear in both diploid and triploid seed tissues. 

To summarize, these results demonstrate that (i) wild barley seed reaches the accumulation of growth at 20–24 DAP; (ii) SMTs and endosperm cells undergo cell death from 6 and 12 DAP, respectively; (iii) endoreduplication is more variable in a mixture of SMTs/embryo. All our observations suggest that wild barley seed ripening and dessication continue after 48 DAP.

## 4. Discussion

Wild barley, *H. vulgare* subsp. *spontaneum* is abundant in diverse ecogeographic regions in the Middle East and has been studied extensively from the phenotypic, genetic, and agronomic perspectives [37,38,39,40]. Wild barley plants growing in habitats with diverse environmental conditions, are exposed to numerous stressors, which directly influence their seed yield [39]. In wild barley, spike and seed traits were expressed so far only by spike length, grain number per spike or TKW [39]. Here, we focused on the size and biomass of dry wild barley kernels. We noted that the ranges of seed traits were much wider in subsp. *spontaneum* than in cultivated barley [11]. Interestingly, dry seeds of wild barleys were on average longer than in the cultivars. It seems that the wild barley seed length might be an interesting trait utilized by the breeders for seed yield improvement *per se*. One desert genotype—MHx deserves special attention. It is characterized by the lowest values of all measured seed features, which might be a result of a region-specific separation [40]. As expected, the seed biomass was higher in the cultivated barley because this is the main yield-related trait used during breeding [39]. The subsp. *spontaneum* accessions had on average longer awns than cultivated barleys, which is in agreement with previous observation [39]. Interestingly all accessions originating from the xeric environments possessed shorter awns, which is an example of an adaptation mechanism adjusting plants to the environment [40].

Until now, advanced methods detecting morphological and cellular changes during seed development have not been used, either from a domestication or stress adaptation point of view, in wild barley. Therefore, we investigated the dynamics of endoreduplication in the diploid and triploid seed tissues from the time shortly after pollination until dry seeds. In parallel, we monitored the morphological and PCD changes accompanying endoreduplication to understand better the complexity of wild barley seed formation (Figure 7). 

There are several parameters for quantification endoreduplication level [43]. Commonly use indicator is cycle value (a.k.a. endoreduplication index) [44]. However, this formula considers 4C (and 6C endosperm) nuclei as already endoreduplicated. Although some 4C nuclei (6C) might already be programmed for endoreduplication, others will be regularly cycling G2 nuclei. FCM does not recognize which 4C nuclei will undergo the mitotic cell cycle and which endocycle. Therefore, we recently introduced a new conservative formula, which considers that 8C nuclei (and 12C nuclei in endosperm) as the first unambiguous level of endoreduplication [11]. 

Many dicots possess non-endospermic seeds, where the developing embryo consumes most of the endosperm before the seed maturation. For the non-endospermic seeds, endoreduplication intensity is a marker of seed quality and maturation [36,45]. In contrast, grasses (*Poaceae*) have endospermic seeds which means that the endosperm forms the major and embryo the minor tissue mass of the fully developed seed. Besides, the high nutritional value of endosperm makes cereals the main crops worldwide to produce energy for humans and livestock. Endoreduplication appears during endosperm development and is correlated with the rapid growth of the caryopsis, the synthesis and accumulation of storage compounds, mainly starch and proteins in cereals [20]. We found that during seed development, both wild and cultivated barley endosperm underwent two rounds of endoreduplication resulting in 12C and 24C nuclei, respectively [11]. Two endocycles also appeared during wheat [46] and rice [47] endosperm development. Four and up to seven rounds of endoreduplication were found in the endosperm of sorghum [48] and maize [49], respectively. This suggests that the upper level of endopolyploidization is genetically regulated in the cereal endoserm, including *H. vulgare*. Further genetic variation most likely exists in the kinetics of endoreduplication which is suggested by the different SCV profiles observed in our study for different genotypes when grown under identical cultivation conditions. In wild barley, the major endoreduplication activity started ~8 DAP, i.e., two days later compared to the cultivated barley [11]. In both taxa, the SCV decreased after 32 DAP. The study performed in cultivated barley has already shown that endosperm endoreduplication nuclei were progressively degraded during the accumulation of the storage materials and ripening. This degeneration was initiated in highly endopolyploid nuclei and accompanied by accumulation of DNA damage and cell death [11]. Interestingly, we detected high proportion of endoreduplicated nuclei in dry seeds for subsp. *spontaneum*. Admittedly, desiccating and dry seeds of cultivated barley also contained endoreduplicated nuclei [11], but not in such proportion as in wild barley. Microscopic observations confirmed that endoreduplicated nuclei originated from AL in dry barley seeds [50]. Endoreduplicated AL nuclei are not observed in other cereals except for barley [50]. 

Here, we also found that wild barley has shifted the major seed/endosperm morphological and developmental phases and needs more time to complete seed ripening comparing to cultivated barley (Figure 6a–c) [11]. Delayed desiccation period and entrance into dough phases were the most obvious differences between wild and cultivated strains [12]. Our findings complement previous observations of several days difference in the heading and anthesis in wild *versus* cultivated barley [39]. Furthermore, the gray color of maturating endosperm in subsp. *spontaneum*, comparing to the white-yellow color of endosperm in cultivated barley [11], may reflect distinct compositions of storage compounds. So far, transcriptomic and metabolomic profiles of seed storage compounds are available for cultivated but not for wild barley [51]. The darker color of wild barley endosperm may also indicate the presence of secondary metabolites, e.g., anthocyanins or other reactive oxygen species scavenging molecules [52,53]. Taken together, all results collected for wild and cultivated barleys raised the question whether there is a link between higher endoreduplication level and different color of endosperm in wild barley. However, solving this question will require further studies, for example examination of the secondary metabolites using high-performance liquid chromatography. Extended analysis may help to better understand the mechanisms of stress adaptation and cereal seed improvement.

Based on the studies in cultivated barley, we concluded that endoreduplication in SMTs is correlated with starch deposition, and in embryos with differentiation of the tissues [11]. We detected two populations of endopolyploid nuclei (8C and 16C reflecting one and two endocycles, respectively) in a mixture of diploid seed tissues of wild barley which is similar to cultivated barley [11]. This is additional evidence suggesting that the number of endocycles is genetically controlled and species-specific [54]. We noted that in the mixture of SMTs/embryo of subsp. *spontaneum* seeds, that endoreduplication peaked two times at 12 and 24 DAP. Comparing wild and cultivated barley, the level of endoreplication expressed by SCV was the same, however, the peaks were shifted. We assign the first endoreduplication peak to SMTs, which correlates with their intensive growth. We assume that the second endoreduplication peak should be attributed to the embryo, and correlate with its rapid growth and tissue differentiation. Similarly to the endosperm, wild barley dry seed tissues contained higher proportion of endoreduplicated nuclei as compared to 48 DAP sample. However, this observation was exclusive only for wild, not cultivated barley (Figure 6). Among ten wild barley accessions, we found variation in the dynamics of SMTs/embryo and endosperm endopolyploidization. On the one hand, some accessions had shifted in time the endoreduplication peak, on the other hand, there were differences with SCV throughout the entire seed development. This finding unravels a new level of variation between wild barley populations. However, it has to be noted that the accessions used in our report represent only a limited diversity and that the variation is probably much greater in wild barley. Intra-specific endpolyoploidy variation is quite common in both cultivated [39,40] and wild [55] plants. Studies performed in Arabidopsis revealed that endoreduplication levels are controlled by the interaction of multiple mostly cell cycle-related genes [55]. 

Importantly, we detected a link between the amount of endoreduplicated nuclei and the ecogeographical origin of the wild barley accessions. Namely, accessions originating from the xeric environments tended to have higher SCV for both SMTs/embryo and endosperm tissues. This is analogous with the previous findings for Israeli accessions of wild barley, in the context of genetic variability detected by molecular markers [56]. Many studies have found endoreduplication more abundantly among plants that grow under environmentally challenging conditions [21,22,27,57]. Increasing DNA content may be integrated into the damage-induced oxidative stress-response systems, like for instance pentose phosphate pathway [58]. In this system, endoreduplication may promote compensation to damages by upregulation of gene expression involved in the overproduction of metabolites [58]. On the other hand, endopolyploidy is thought to play significant roles in plant physiology [21]. Altered phytohormone balances, changed after exposition to environmental stressors, probably trigger organ-specific endopolyploidization [24]. This may suggest an adaptive mechanism to an environmental gradient that results in differential endopolyploidy [24]. With only ten accessions used in this study, the identification of an obvious adaptive response to harsh environmental conditions is not conclusive. Therefore, to identify a potential link between environmental gradient and seed endoreduplication variation, future studies involving a larger number of genetically defined samples and mapping causal genes are necessary. 

## 5. Conclusions and Future Perspectives

Both diploid and triploid barley seed tissues undergo two endocycles. This study of endoreduplication in wild barley seeds revealed a new level of variation appearing within subsp. *spontaneum*. Wild barley had a higher endoreduplication level in endosperm tissues comparing with the with cultivated one and the amount of endoreduplicated nuclei tended to be higher in xeric accessions. We are currently aiming to better understand how spatiotemporal seed endoreduplication patterns change under various stresses and whether these stresses are linked to stress adaptation.

## Figures and Tables

**Figure 1 genes-12-00711-f001:**
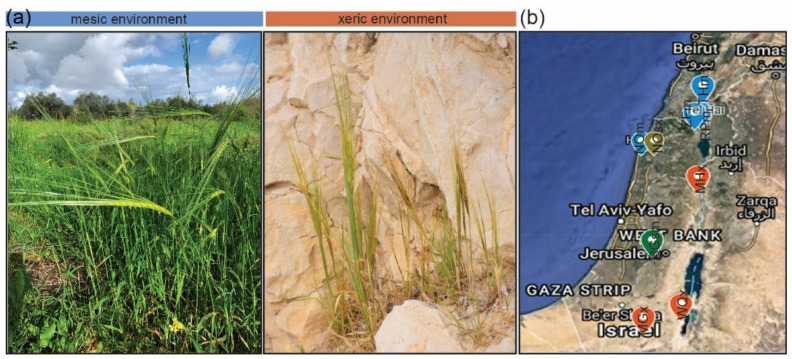
Geographic origin of wild barley (*H. vulgare* subsp. *spontaneum*) accessions. (**a**) Examples of wild barley from the Galilee (mesic) and Judean desert (xeric) in Israel. (**b**) Collection sites in Israel. Blue points = mesic sites (THm = Tel Hai, ZFm = Zefat, RPm = Rosh Pinna, NOm = Nahal Oren northern facing slope); green point = semi-mesic site (BGsm = Bar Giyyora); green-brown point = semi-xeric site (NOsx = Nahal Oren southern facing slope); orange points = xeric sites (MHx = Mehola, MGx = Machtesh Gadol, WQx = Wadi Qilt). The map was generated using Google Maps. Detailed ecogeographical data are presented in Table 1.

**Figure 2 genes-12-00711-f002:**
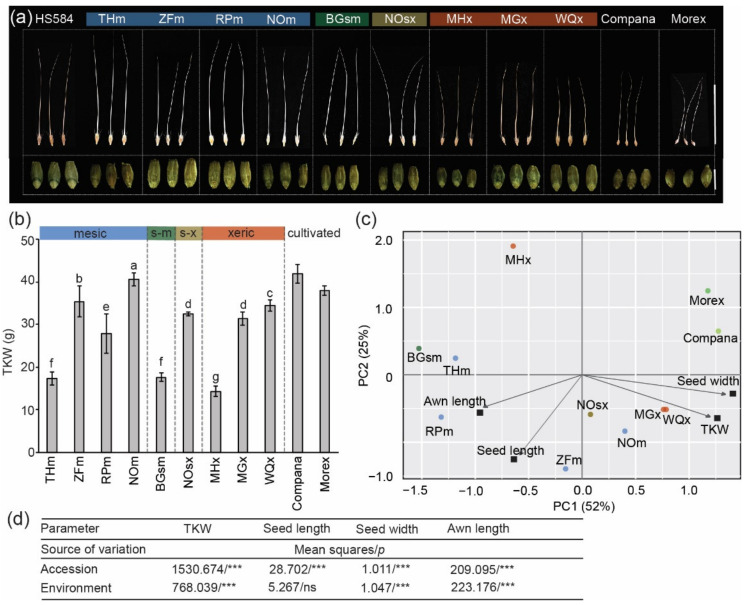
Dry seed phenotypes of wild barley accessions. (**a**) Dorsal views of hulled dry seeds with awns (upper panel; Scale bar = 10 cm) and peeled dry seeds (lower panel; scale bar = 10 mm). Mesic accessions: THm = Tel Hai, ZFm = Zefat, RPm = Rosh Pinna, NOm = Nahal Oren NSF; semi-mesic accession: BGsm = Bar Giyyora; semi-xeric accession: NOsx = Nahal Oren SFS; xeric accessions: MHx = Mehola, MGx = Machtesh Gadol, WQx = Wadi Qilt. All genotypes were grown in phytochamber under the same conditions. (**b**) Quantitative data for the thousand-kernel weight (TKW). Data are the means (±SD) from three biological replicates. Values marked with the same letter do not differ according to Duncan multiple range tests (*p* ≤ 0.05). (**c**) Principal component (PC) analysis of TKW, seed length, and width for peeled seeds, and awn length. The positions represent contribution rates of the two PCs (Source data are shown in Appendix A, other combinations of PCs are presented in Appendix A). The ecological conditions at the sampling site of HS584 are unknown. Compana and Morex represent two- and six-rowed cultivated barley controls, respectively. (**d**) Summary of ANOVA performed for seed traits. The sources of variance were as follows: nine accessions and four environment types. *** Significant at *p* ≤ 0.001; ns—not significant.

**Figure 3 genes-12-00711-f003:**
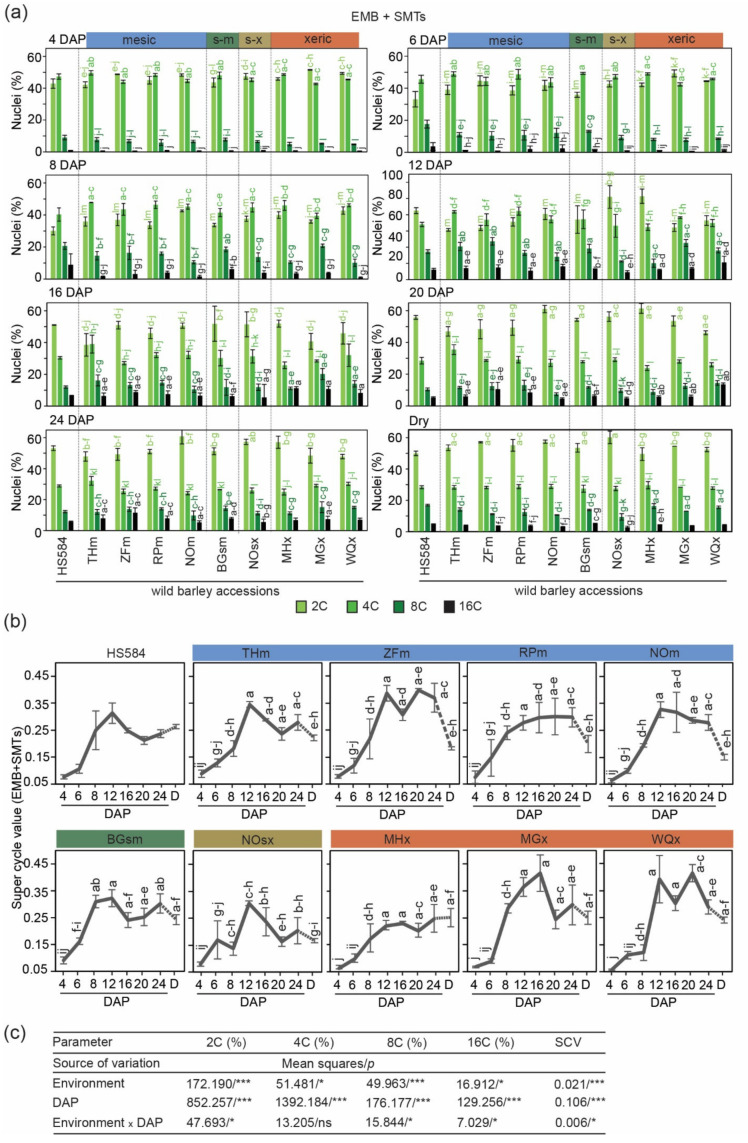
Estimation of C-values in diploid seed tissues represented by the embryo (EMB) and seed maternal tissues (SMTs) of ten wild barley accessions originating from Israel. (**a**) Percentage of 2C, 4C, 8C and 16C nuclei at a given day after pollination (DAP) and in dry seeds. Mesic accessions: THm = Tel Hai, ZFm = Zefat, RPm = Rosh Pinna, NOm = Nahal Oren NSF; semi-mesic accession: BGsm = Bar Giyyora; semi-xeric accession: NOsx = Nahal Oren SFS; xeric accessions: MHx = Mehola, MGx = Machtesh Gadol, WQx = Wadi Qilt. The ecological conditions at the sampling site of HS584 are unknown. Data are the means (±SD) from three biological replicates, each with at least 5 individual measurements (seeds). Data marked with the same letter do not differ according to the Duncan test (*p* ≤ 0.05) (Source data are shown in Appendix A (**b**) Super cycle values at a given DAP calculated based on the data from (**a**), D = dry seed. The dashed line between 24 DAP and dry seed samples indicates further seed development after 24 DAP (Source data are shown in Appendix A) (**c**) Summary of ANOVA performed for (**a**) and (**b**). The sources of variance were as follows: four environment types, eight-time point (DAP), and interaction between environment and DAP. *, *** Significant at *p* ≤ 0.05, 0.001, respectively; ns—not significant.

**Figure 4 genes-12-00711-f004:**
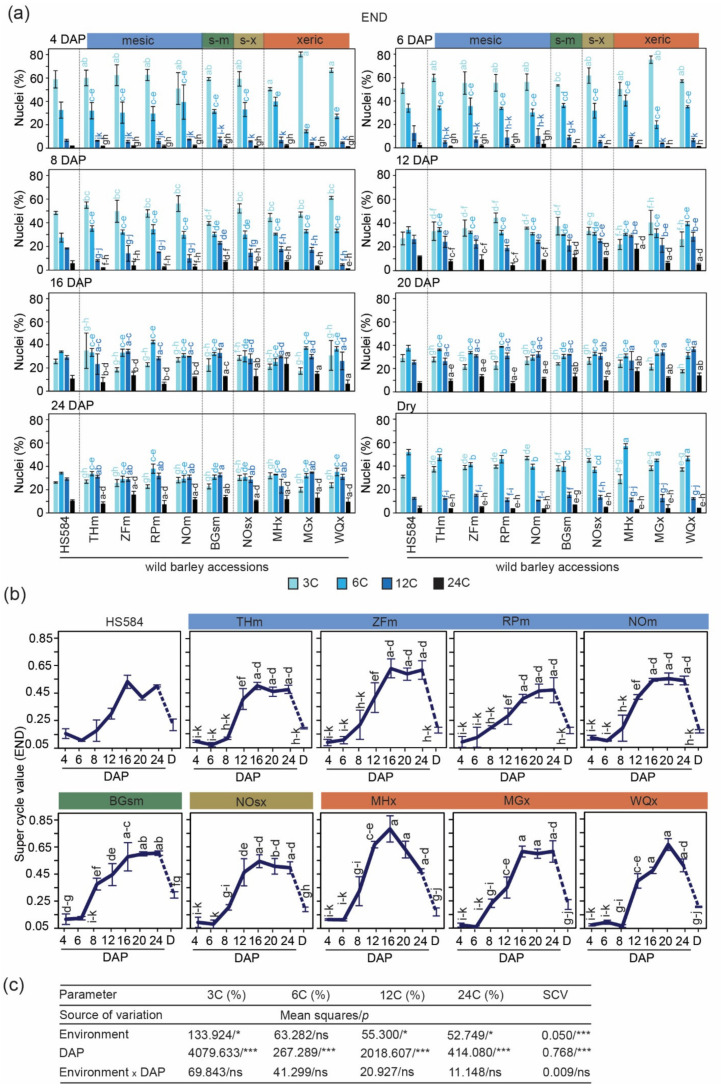
Estimation of C-values in triploid endosperm (END) tissues of ten analyzed wild barley accessions. (**a**) Percentage of 3C, 6C, 12C and 32C nuclei at a given day after pollination (DAP) and in dry seeds. Mesic accessions: THm = Tel Hai, ZFm = Zefat, RPm = Rosh Pinna, NOm = Nahal Oren NSF; semi-mesic accession: BGsm = Bar Giyyora; semi-xeric accession: NOsx = Nahal Oren SFS; xeric accessions: MHx = Mehola, MGx = Machtesh Gadol, WQx = Wadi Qilt. The ecological conditions at the sampling site of HS584 are unknown. Data are the means (±SD) from three biological replicates, each with at least 5 individual measurements (seeds). Data marked with the same letter do not differ according to the Duncan test (*p* ≤ 0.05) (Source data are shown in Appendix A) (**b**) Super cycle values at a given DAP calculated based on the data from (**a**), D = dry seed. The dashed line between 24 DAP and dry seed samples indicates further seed development after 24 DAP (Source data are shown in Appendix A). (**c**) Summary of ANOVA performed for (**a**) and (**b**). The sources of variance were as follows: four environment types, eight-time point (DAP), and interaction between environment and DAP. *, *** Significant at *p* ≤ 0.05, 0.001, respectively; ns—not significant.

**Figure 5 genes-12-00711-f005:**
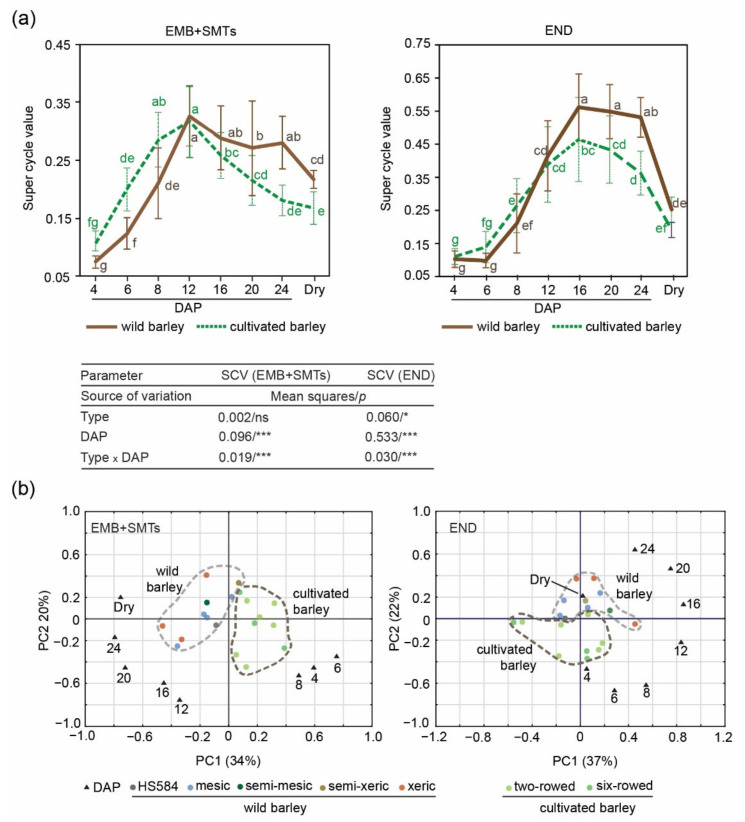
Comparison of the super cycle values (SCVs) for diploid seed tissues (EMB + SMTs) versus triploid endosperm (END) at given DAP between wild and cultivated barley. (**a**) Mean SCVs of EMB + SMTs and endosperm. Wild barley is represented by ten accessions (solid brown lines). Cultivated barley includes six two-rowed and three six-rowed cultivars (dashed green lines). Data for six-rowed cultivars are presented in Appendix A. Data are the means (±SD) from three biological replicates, each with at least 5 individual measurements (Source data are presented in Appendix A). Data marked with the same letter do not differ according to the Duncan test (*p* ≤ 0.05). The sources of variance were as follows: two types of barley (wild and cultivated), eight-time point (DAP), and interaction between type and DAP. *, *** Significant at *p* ≤ 0.05, 0.001, respectively; ns—not significant. (**b**) Principal component (PC) analysis of SCVs in wild and cultivated barley. Numbers in the plots indicate DAP. The positions represent the contribution rates of the two main PCs to a given character. The dashed-line areas were added to highlight sample similarity.

**Figure 6 genes-12-00711-f006:**
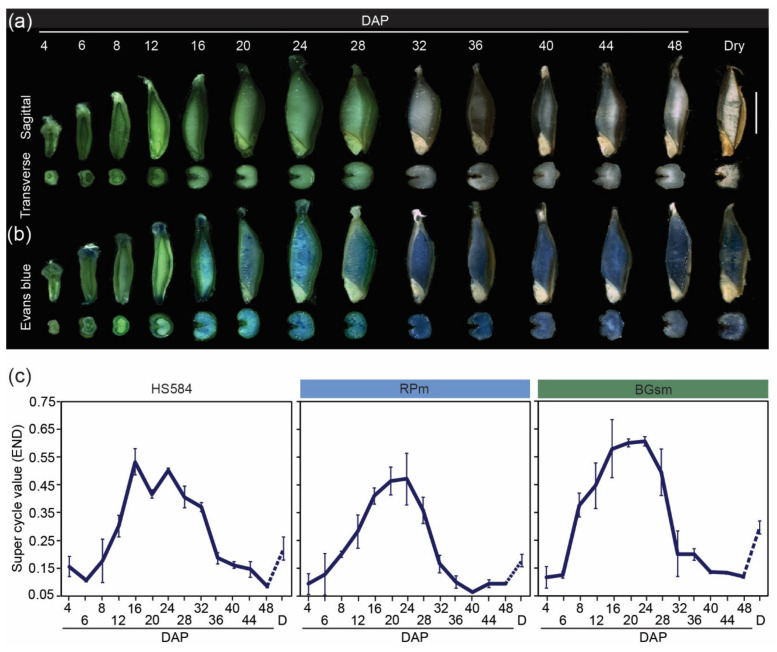
Time-course study of morphological and cellular events in developing wild barley seeds from 4 to 48 days after pollination (DAP). (**a**,**b**) Representative seed sections of HS584 (**a**) without and (**b**) with 0.1% Evans blue staining. The seeds shown are representative of at least 20 individuals not stained (**a**) and stained seed (**b**). Scale bar = 5 mm. (**c**) Endosperm super cycle values of three wild barley accessions: HS584, Rosh Pinna and Bar Giyyora. Complementary data for diploid seed tissues are presented in Appendix A. Data are the means (±SD) from three biological replicates, each with at least 5 individual measurements (seeds). The dashed line between 48 DAP and dry seeds (D) samples represents the desiccation stage that was not analyzed in detail here.

**Figure 7 genes-12-00711-f007:**
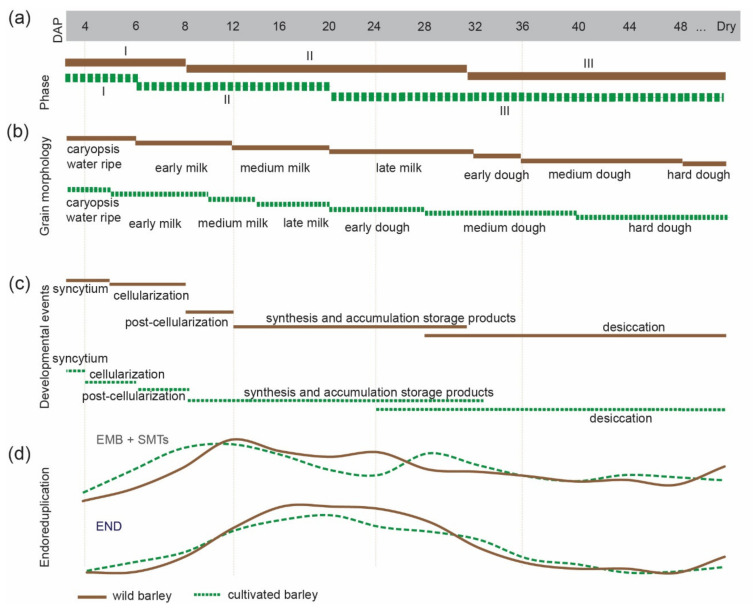
The model of phase transitions during wild (brown, solid lines) and cultivated barley [11] (green, dashed lines) seed development. Overview of (**a**) phase of barley grain development (based on [20,41], and (**b**) grain morphological changes (based on [12]), (**c**) developmental events (based on [20,41,42] and (**d**) endoreduplication dynamics. EMB + SMTs = embryo and seed maternal tissues and END = endosperm.

**Table 1 genes-12-00711-t001:** Sampling sites and ecogeographical data of the analyzed wild barley accessions.

Locality/Name (All within Israel)	Genebank	Acronym	Type of Environment	Longitude (N)	Latitude (E)	Altitude (a.s.l.)	Maximum Temperature in April (°C)	Rainfall in April (mm)	Annual Rainfall (mm)	Average Annual Humidity at 14:00 Mean ± SD	Annual Evaporation (cm)	Soil Type
Bar Giyyora	ICGB	BGsm	Semi-mesic	35.083333	31.716667	760	22	18	535	47.1 ± 10.8	215	T
HS584	IPK	HS584	n.a.	n.a.	n.a.	n.a.	n.a.	n.a.	n.a.	n.a.	n.a.	n.a.
Machtesh Gadol	ICGB	MGx	Xeric	35.000000	30.950000	n.a.	25	3	70	n.a.	n.a.	n.a.
Mehola	ICGB	MHx	Xeric	35.533333	32.350000	−150	30	6	<200	37.5	240	A
Nahal Oren	ICGB	NOm	Mesic	34.966667	32.716667	n. a.	24	13	584	n.a.	n.a.	n.a.
Nahal Oren	ICGB	NOsx	Semi- xeric	34.966667	32.716667	n. a.	24	13	584	n.a.	n.a.	n.a.
Rosh Pinna	ICGB	RPm	Mesic	35.550000	32.983333	700	25	20	535	43.6 ± 10.5	220	T
Tel Hai	ICGB	THm	Mesic	35.573979	33.234719	400	26	23	768	46.9 ± 7.6	220	T
Wadi Qilt	ICGB	WQx	Xeric	35.44565	31.859	50	30	6	<200	34.7 ± 9.3	330	A
Zefat	ICGB	ZFm	Mesic	35.496001	32.969206	800	20	27	670	50.4 ± 13.1	220	R

This table was partially prepared based on [35] and https://ims.gov.il/he/ClimateAtlas, accessed on 15 March 2021. ICGB = Institute of Evolution Wild Cereal Gene Bank at the University of Haifa, Israel; IPK = Leibniz Institute of Plant Genetics and Crop Plant Research, Gatersleben, Germany. Annual rainfall = average for the period 1981–2010. Average maximal temperature and rainfall in April were recorded in 1995–2009. a.s.l.—above sea level; n.a.—not available; SD—standard deviation over mean monthly data; Soil type: A = alluvium, R = rendzina, T = terra rossa.

## Data Availability

Not applicable.

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
