# Peer review of "Endopolyploidy Variation in Wild Barley Seeds across Environmental Gradients in Israel"

_genes, 2021, doi:10.3390/genes12050711_

Round 1

Reviewer 1 Report

General comments:

In the manuscript “Endopolyploidy Variation in Wild Barley Seeds across Environmental Gradients in Israel”, the authors describe variation in endoreduplication of barley seeds from different habitats and wild vs tame plants. The introduction, methods, and results are well-written and easy to understand, however the discussion would benefit from a thorough edit to improve some grammatical errors and awkward wording. Generally the methodology seems sound, although some clarification of replicates would be appreciated. The results section is overly long with a lot of information repeated in the figures and in the text – if it could be simplified and streamlined it would be easier to follow. While there is a considerable amount of strong data, the discussion needs to be strengthened to demonstrate the importance of these results (for example see comments below on lack of discussion on morphological variation in seeds). I think this paper could be valuable and presents an interesting area for further research, with some adjustments to the discussion and interpretation of results.

Specific comments:

Lines 57-77: while not critical, it might be nice to see some simple diagrams in this section as visuals to accompany the text?

Line 62: “into the dormancy” remove “the”

Lines 99-104: This may be a point of personal preference, but I would not include the major findings in the introduction. I would use these lines to provide a brief overview of how you tested your objective (provided in lines 97 and 98).

Line 116: abbreviation “NSF” should be “NFS”

Line 144: how were seeds dried? Any special conditions?

Line 154: please expand on somatic tissue control – do 2-month old leaves contain any endoreplicated nuclei or are they predominantly in 2C? (I think you explain this in the results at line 212-213 but should be included in methods).

Line 155: “plants were served as” remove “were”

Line 158: expand on what “peeled seeds” means (will not be clear for some readers)

Line 169: can you explain how you know that all of the 4C nuclei represent nuclei in G2 of the cell cycle and are not also endopolyploid? Is there a standard ratio of how many cells are undergoing the cell cycle in a seed at a given time? In a lot of plant endopolyploidy literature  (using leaf tissue) 4C would be the first endopolyploid level.

Line 178: idea of replicates not clearly defined to this point - do you mean 20 seeds per individual plant parent? For example at line 129, how many grains were germinated and how many adult plants grew to seed and were harvested? And at line 142 you indicate 10 grains were collected from each plant – was this just for flow cytometry (at line 156 you state you used “at least 5 different seeds”).

Line 181: “hulled seeds possessed awns” should say “hulled seeds that possessed awns”

Line 185: for those that don’t know, you need to introduce this section to say why/what Evan’s blue staining is.

Line 208: list with smaller number first

Line 225: “nuclei, oscillating” include “each” before oscillating

Lines 232-242: I found it difficult to interpret the heat maps in the supplementary file. For example, is figure S4(b) and S5(b) the only example of where the heat map indicates something because there is a pattern? There are many different uses of heat maps so a bit more explanation might be required.

Figure 1: could figure 1(a) and (b) come earlier in the manuscript? Would break up the text nicely.

Figure 2: could include color-coding legend here again (for mesic, xeric, etc.)

Lines 286-325: I found this section a bit confusing – it could benefit from paring down the information or better use of tables & figures (figure 3).

Line 288: Use of phrase “analogously to” here (and at lines 301, 309, 358, 482) is awkward. I think a better phrase might be “similar to”.

Line 399: “changed the color from” remove “the”

Line 470: “endoreduplication” should be “endoreduplicated”

Lines 474-476: sentence needs to be reworded. Is there other literature or evidence to explain the color change of the seeds?

Lines 477-479: explain why this is important to understand

Line 480: correct to “endopolyploidization” or “endoreduplication”

Line 481: “with” accidentally written twice

Line 482: “analogously”

Line 484: “another evidence” should be “more evidence” or “additional evidence”

Line 494: stay specific throughout – “endoploidization” should be “endopolyploidization”?

Line 495: “reached the endopolyploidy peaks earlier” – do you mean to say that endopolyploid nuclei were observed earlier in time?

Line 496: “Evidence of genetic variation” – be clear about how you can come to this conclusion. You may want to link back to the statistical findings to indicate where there was a significant difference between accessions. Comparisons can be made between wild vs. cultivars perhaps (can you say if there is any genetic information or seed “pedigree” information to support the relatedness of the cultivars?) Figure 1c alone shows tremendous variation (e.g., awn length and seed size) between the xeric, mesic, and cultivated samples, but this is not included in the discussion.

Line 497: remove “besides”

Line 502: “these” should be “those”

Lines 513, 519: using citation “21, and references therein” is not a bad idea, but it would be helpful to have more direct citations to 21 as well as the original literature if it clearly links to your statement.

Line 529: start sentence with “This endoreduplication”

Line 534: climate change seems to be randomly thrown in here. Link it back to your findings with mesic and xeric environments, potential stress response, etc..

Supplementary Figure S1(c): typo in y axis label “length”

Author Response

Reviewer#1

General comments:

In the manuscript “Endopolyploidy Variation in Wild Barley Seeds across Environmental Gradients in Israel”, the authors describe variation in endoreduplication of barley seeds from different habitats and wild vs tame plants. The introduction, methods, and results are well-written and easy to understand, however the discussion would benefit from a thorough edit to improve some grammatical errors and awkward wording. Generally the methodology seems sound, although some clarification of replicates would be appreciated. The results section is overly long with a lot of information repeated in the figures and in the text – if it could be simplified and streamlined it would be easier to follow. While there is a considerable amount of strong data, the discussion needs to be strengthened to demonstrate the importance of these results (for example see comments below on lack of discussion on morphological variation in seeds). I think this paper could be valuable and presents an interesting area for further research, with some adjustments to the discussion and interpretation of results.

Response: Thank you for the valuable comments. Based on the new statistics we rewrote the Results and improved the Discussion chapters.  In addition, we included more details about methodology (Material and method section).

Specific comments:

Lines 57-77: while not critical, it might be nice to see some simple diagrams in this section as visuals to accompany the text?

Response: We included this information in Fig. 6a.

Line 62: “into the dormancy” remove “the”

Response: The sentence was corrected.

Lines 99-104: This may be a point of personal preference, but I would not include the major findings in the introduction. I would use these lines to provide a brief overview of how you tested your objective (provided in lines 97 and 98).

Response: The paragraph was rewritten. We added the information about applied methods and underlined the objectives of the paper. The new version contains objectives and methods.  

Line 116: abbreviation “NSF” should be “NFS”

Response: The sentence was corrected.

Line 144: how were seeds dried? Any special conditions?

Response: Detailed information about collecting and storing dry seeds was put at the end of Subsection: 2.1. Plant materials and growth conditions. We collected the mature dry seed at the hard-dough phase according to the barley grain development scale of Tottman et al., 1987, An explanation of the decimal code for the growth stages of cereals, with illustrations.  Following the morphological scale of barley/endosperm formation shown in this paper, measuring the water content is not necessary.

Line 154: please expand on somatic tissue control – do 2-month old leaves contain any endoreplicated nuclei or are they predominantly in 2C? (I think you explain this in the results at line 212-213 but should be included in methods).

Response: To avoid misunderstanding, we excluded the information about leaves.

Line 155: “plants were served as” remove “were”

Response: A new version of the manuscript does not contain this paragraph.

Line 158: expand on what “peeled seeds” means (will not be clear for some readers)

Response: Peeled means with removed hulls. Hulls cover and protect the barley seed.  From an anatomical point of view, they consist of lemma, palea, and glumes.  We added the information about hulls in the Introduction (paragraph describing seed anatomy)

Line 169: can you explain how you know that all of the 4C nuclei represent nuclei in G2 of the cell cycle and are not also endopolyploid? Is there a standard ratio of how many cells are undergoing the cell cycle in a seed at a given time? In a lot of plant endopolyploidy literature  (using leaf tissue) 4C would be the first endopolyploid level.

Response:  Some 4C nuclei might already be programmed for endoreduplication, others will not be. However, using flow FCM it is not possible to distinguish which cell will undergo mitotic cell cycle, and which are already programmed for the endocycle. Most of the literature is inclusive in this aspect (considers also an unknown fraction of G2 4C nuclei as endoreduplicated). We took a more conservative approach here by using the super cycle value. Though we lose some endoreduplication primed 4C (or 6C nuclei in endosperm), we do not “contaminate” our calculation with the mitotically cycling nuclei. We referred to this problem in the Discussion.

Line 178: idea of replicates not clearly defined to this point - do you mean 20 seeds per individual plant parent? For example at line 129, how many grains were germinated and how many adult plants grew to seed and were harvested? And at line 142 you indicate 10 grains were collected from each plant – was this just for flow cytometry (at line 156 you state you used “at least 5 different seeds”).

Response: For each accession, we have grown five plants. This information we added into subsection 2.1. Plant materials and growth conditions. In addition, each subsection of Materials and methods describing used methods was supplemented by information about the number of used seeds, and biological replicates. This information can be found also in the description of the figures.

Line 181: “hulled seeds possessed awns” should say “hulled seeds that possessed awns”

Response: The sentence was corrected.

Line 185: for those that don’t know, you need to introduce this section to say why/what Evan’s blue staining is.

Response: We specified the title of this subsection from “Evans blue staining assay” to “Cell death assay by Evans blue staining”. In addition, we put the information about PCD into the Introduction. In the subsection of results 3.4. Morphological and cellular changes during 48 days of wild barley seed development, we explained how Evans blue works.

Line 208: list with smaller number first

Response: The sentence was corrected.

Line 225: “nuclei, oscillating” include “each” before oscillating

Response: The sentence was corrected.

Lines 232-242: I found it difficult to interpret the heat maps in the supplementary file. For example, is figure S4(b) and S5(b) the only example of where the heat map indicates something because there is a pattern? There are many different uses of heat maps so a bit more explanation might be required.

Response: After re-doing the statistics, the heat maps were removed.

Figure 1: could figure 1(a) and (b) come earlier in the manuscript? Would break up the text nicely.

Response: We divided Figure 1 into two smaller. The new Figure 1 was put into subsection 2.1. Plant materials and growth conditions.

Figure 2: could include color-coding legend here again (for mesic, xeric, etc.)

Response: Both Figures 3 and 4 (old Figures 2 and 3, respectively) contain color-coding legend on the top and bottom parts.

Lines 286-325: I found this section a bit confusing – it could benefit from paring down the information or better use of tables & figures (figure 3).

Response:  This paragraph was shortened and simplified.

Line 288: The use of the phrase “analogously to” here (and at lines 301, 309, 358, 482) is awkward. I think a better phrase might be “similar to”.

Response: This was corrected.

Line 399: “changed the color from” remove “the”

Line 470: “endoreduplication” should be “endoreduplicated”

Response: Both sentences were corrected.

Lines 474-476: sentence needs to be reworded. Is there other literature or evidence to explain the color change of the seeds?

Response: We found that the gray color of wild barley endosperm may indicate the presence of anthocyanin. We introduced this topic to the discussion.

Lines 477-479: explain why this is important to understand

Response: We expand this topic/paragraph.

Line 480: correct to “endopolyploidization” or “endoreduplication”

Line 481: “with” accidentally written twice

Line 482: “analogously”

Line 484: “another evidence” should be “more evidence” or “additional evidence”

Line 494: stay specific throughout – “endoploidization” should be “endopolyploidization”?

Response: All sentences were corrected.

Line 495: “reached the endopolyploidy peaks earlier” – do you mean to say that endopolyploid nuclei were observed earlier in time?

Response: The sentence was written in another way.

Line 496: “Evidence of genetic variation” – be clear about how you can come to this conclusion. You may want to link back to the statistical findings to indicate where there was a significant difference between accessions. Comparisons can be made between wild vs. cultivars perhaps (can you say if there is any genetic information or seed “pedigree” information to support the relatedness of the cultivars?) Figure 1c alone shows a tremendous variation (e.g., awn length and seed size) between the xeric, mesic, and cultivated samples, but this is not included in the discussion.

Response: We mitigated the statements/conclusions about the genetic variation of endopolyploidization and we left only the statement about variation. 

Line 497: remove “besides”

Line 502: “these” should be “those”

Response: Both sentences were corrected.

Lines 513, 519: using citation “21, and references therein” is not a bad idea, but it would be helpful to have more direct citations to 21 as well as the original literature if it clearly links to your statement.

Response: The references were corrected.

Line 529: start sentence with “This endoreduplication”

Response: The sentence was corrected.

Line 534: climate change seems to be randomly thrown in here. Link it back to your findings with mesic and xeric environments, potential stress response, etc..

Response: The chapter Conclusion was improved.

Supplementary Figure S1(c): typo in y axis label “length”

Response: The typing error was corrected.

Reviewer 2 Report

This is interesting paper showing variation in endopolyploidy level in barley seeds originating from different sites in Israel. I consider the experiment to be very good and its results worth of publishing. The introduction and methods are very good. The major problem I see in the statistics which is rather poor and not adequate to the complexity of the presented experiment. The results so need to be explained in a complicated fashion and are not very readable, understandable and always believable. The discussion seem be a bit brief given the complexity of the results and the topic at all.

Specific comments:
The paper use only very basic statistics which seem me not much adequate to the complexity of conducted experiment. The repeated ANOVa tests done for each seed age category for the same groups seeds suffers from multiple comparison error (e.g. https://en.wikipedia.org/wiki/Multiple_comparisons_problem) and the probability inferences can be biased in this case. Here I would rather see some statistical model with seed age, locality, and environment used as explanatory factors (simultaneously in the model). With such a kind of analysis the results could be more easily understand and this section could be much reduced.

l. 509-510 How this variability was formally tested - I saw no tests of variability in the paper. It seems that authors deduce about the extent of this variability from observing changes of p-level from the conducted ANOVA tests. This is nearly meaningless. The significance tells about differences among groups, assuming there is similar variability within groups. But have really all groups the same variances – even to can calculate ANOVA here? Even if all will be O.K. with ANOVA assumptions highly significant ANOVA with p<0.001 can mean that samples differ 10-times, 50-time or even more and highly significant p need not to be much helpful here in answering which one of variable groups is more variable that the other. Here authors might find useful to transform their data to a “variance” measure by susbtracting original values from their total mean and to repeat the ANOVA or other analyses with absolute values of these numbers. If a sample will show these values significantly higher than the other sample, it will have significantly higher variation.

Fig. 1e: It seems suspicious to me that in the PCA analysis of TKW, seed length and seed width (i.e. the three variables), the first two PC axes explains in sum only 44% of variation. In theory they should explain nearly everything with the last 3rd PC axis explaining the rest to the 100%.

l. 464, 484. It is not clear to me how genetic regulation of this process can be deduced from the mentioned observations (applies also to l. 528 of the conclusions).

l. 474: I am not sure if this diference was tested anywhere in the paper? Was it and how?

Fig. 6: I lack information, how the model presented in this figure was constructed. Explain this.

I am a bit curious from the fact that endopolpyloidy can decrease in time which is very unusual for other tissue types, such as for leaves. Could authors spend some space by discussing this in more detail in their paper. It is clear that this can perhaps happen by only by reduction of cell numbers. However, the magnitude of this change is really surprising for me.  From Fig. 5 c and others showing the temporal trend in super cycle value change it seems that supercycle values can reach the same values as at the beginning of seed initiation after 48 days – meaning that fast all endopolyploid cells appearing during seed development are later destroyed. How? Where the nuclei disappear? And is this truly endopolpyloidy and not some peculiar form of normal polyploidy and cell cycle, where the multiplied genome copies are later distributed to new cells – which will explain this very elegantly, at least from the mathematical point of view?

There is lot of discussion about changes in proportions of 8C, 16C, etc. nuclei in the results but no statistical test showing whether this proportional changes are significant or not. This applies also to the many graphs (e.g. Fig. 2a) where one is not sure if the accession differ or not in this parameter (this is not possible to deduce from the figure as one do not see the variability of the results).

I recomend not to use terms such as C-values and Nuclear DNA content in the paper as this is commonly used in studies of genome size and this can be confusing for many readers (expecting to see some values in picograms). Instead, could you tell you measured “ploidy” of your nuclei or seeds?

First paragraph of the discussion repeats mostly (unnecessarily) the introduction and the methods.   

Author Response

Reviewer#2

This is interesting paper showing variation in endopolyploidy level in barley seeds originating from different sites in Israel. I consider the experiment to be very good and its results worth of publishing. The introduction and methods are very good. The major problem I see in the statistics which is rather poor and not adequate to the complexity of the presented experiment. The results so need to be explained in a complicated fashion and are not very readable, understandable and always believable. The discussion seem be a bit brief given the complexity of the results and the topic at all.

Specific comments:
The paper use only very basic statistics which seem me not much adequate to the complexity of conducted experiment. The repeated ANOVa tests done for each seed age category for the same groups seeds suffers from multiple comparison error (e.g. https://en.wikipedia.org/wiki/Multiple_comparisons_problem) and the probability inferences can be biased in this case. Here I would rather see some statistical model with seed age, locality, and environment used as explanatory factors (simultaneously in the model). With such a kind of analysis the results could be more easily understand and this section could be much reduced.

Response: Thank you for your very constructive comments. We performed two-factors ANOVA. As factors, we used type of environment (xeric, semi-xeric, semi-mesic and mesic) and the day after pollination (from 4 to 24 DAP).  The third factor (locality) was unique and it was not possible to include it in this analysis. We continued the analysis with Duncan’s, post hoc test. We changed Figures 3 and 4 (former Figures 2 and 3, respectively) by presenting the proportion of nuclei for individual C-values (a) as separate columns with SD. In addition, both parts (a) and (b) contain letters being the output of the post hoc test. We also put as point (c) the tables with the output of ANOVA, the tables contain the Source of variation, Mean squares values with marked significance at p=0.05, and p=0.001. In addition, source data we show in the Supplementary file (Tables 1-5).

After changing the statistical analysis, we also rewrote the Results and Discussion.

  1. 509-510 How this variability was formally tested - I saw no tests of variability in the paper. It seems that authors deduce about the extent of this variability from observing changes of p-level from the conducted ANOVA tests. This is nearly meaningless. The significance tells about differences among groups, assuming there is similar variability within groups. But have really all groups the same variances – even to can calculate ANOVA here? Even if all will be O.K. with ANOVA assumptions highly significant ANOVA with p<0.001 can mean that samples differ 10-times, 50-time or even more and highly significant p need not to be much helpful here in answering which one of variable groups is more variable that the other. Here authors might find useful to transform their data to a “variance” measure by susbtracting original values from their total mean and to repeat the ANOVA or other analyses with absolute values of these numbers. If a sample will show these values significantly higher than the other sample, it will have significantly higher variation.

Response: We removed the heat maps of ANOVA from the Supplementary part, instead of this we show there all mean raw data ±SD. In addition, we rewrote these ambiguous sentences in the Discussion part.

Fig. 1e: It seems suspicious to me that in the PCA analysis of TKW, seed length and seed width (i.e. the three variables), the first two PC axes explains in sum only 44% of variation. In theory they should explain nearly everything with the last 3rd PC axis explaining the rest to the 100%.

Response: We included also the fourth variable, i.e. awn length, to PCA analysis and repeated it. PC1 = 52%, PC2 = 25%, PC3 = 18%. Data for PC1 and PC2 are presented in Figure 2, the other two combinations are shown in the Supplementary part.

  1. 464, 484. It is not clear to me how genetic regulation of this process can be deduced from the mentioned observations (applies also to l. 528 of the conclusions).

Response: Data for other species and tissues show that the number of endocycles is often species-specific (e.g. paper by Chevalier et al. Elucidating the functional role of endoreduplication in tomato fruit development. Ann. Bot. 2011) and under the control of several genes. In addition, studies in Arabidopsis show that endoreduplication level is controlled by the interaction of multiple QTLs (Sterken et al., Combined linkage and association mapping reveals CYCD5;1 as a quantitative trait gene for endoreduplication in Arabidopsis. Proc. Natl. Acad. Sci. U. S. A. 2012).

  1. 474: I am not sure if this diference was tested anywhere in the paper? Was it and how?

Response: We found that in wild barley endosperm has a different color, than in cultivated barley. In our previous paper (Nowicka et al. 2021, Dynamics of endoreduplication in developing barley seeds, https://academic.oup.com/jxb/article-abstract/72/2/268/5917133?redirectedFrom=fulltext), we present the data for cultivated barley.  In the Discussion, we added references to this paper. Either in our previous work about cultivated barley or in wild barley, we didn’t measure the composition of endosperm (starch, proteins, etc). We are more focused on cytological and molecular studies.

Fig. 6: I lack information, how the model presented in this figure was constructed. Explain this.

Response: Model compares finding for wild barley (current paper) with cultivated barley (our previous paper, Nowicka et al. 2021, Dynamics of endoreduplication in developing barley seeds,).  For better interpretation, we changed the Figure description.

I am a bit curious from the fact that endopolpyloidy can decrease in time which is very unusual for other tissue types, such as for leaves. Could authors spend some space by discussing this in more detail in their paper. It is clear that this can perhaps happen by only by reduction of cell numbers. However, the magnitude of this change is really surprising for me.  From Fig. 5 c and others showing the temporal trend in super cycle value change it seems that supercycle values can reach the same values as at the beginning of seed initiation after 48 days – meaning that fast all endopolyploid cells appearing during seed development are later destroyed. How? Where the nuclei disappear? And is this truly endopolpyloidy and not some peculiar form of normal polyploidy and cell cycle, where the multiplied genome copies are later distributed to new cells – which will explain this very elegantly, at least from the mathematical point of view?

Response: As shown in our previous study in cultivated barley, endoreduplicated endosperm and SMTs nuclei accumulate DNA damage and are eliminated. Hence, there is a link between endoreduplication and programmed cell death during barley grain and generally cereal development. The program cell death does not affect aleurone layer cells that appear later during endosperm development. So, there is the replacement of endoreduplicated nuclei from the central starchy endosperm which are simply degraded by the lower ploidy aleurone layer nuclei. This causes a reduction in SCV. We discussed this topic in our paper about cultivated barley, here we only mention, why endoreduplicated nuclei die.

There is lot of discussion about changes in proportions of 8C, 16C, etc. nuclei in the results but no statistical test showing whether this proportional changes are significant or not. This applies also to the many graphs (e.g. Fig. 2a) where one is not sure if the accession differ or not in this parameter (this is not possible to deduce from the figure as one do not see the variability of the results).

Response: We changed the way of data presentation. Now, we show the data not as stacked columns, but as clustered column charts. The new graph contains also SD, and the latter being the output of the post hoc test. In addition, raw data are available in the Supplementary part.

I recomend not to use terms such as C-values and Nuclear DNA content in the paper as this is commonly used in studies of genome size and this can be confusing for many readers (expecting to see some values in picograms). Instead, could you tell you measured “ploidy” of your nuclei or seeds?

Response: We discussed this terminology in length internally and also with the expert in the field, prof. Jaroslav Dolezel. The conclusion was that the terms C-values and nuclear DNA are the most appropriate. That terminology we used in our previous paper (Nowicka et al. 2021, Dynamics of endoreduplication in developing barley seeds). The term “ploidy” would be confusing as we talk all the time about diploid (or triploid endosperm) nuclei with varying numbers of sister chromatids.

First paragraph of the discussion repeats mostly (unnecessarily) the introduction and the methods.   

Response: The discussion was rewritten.